# Chloroplast Functionality at the Interface of Growth, Defense, and Genetic Innovation: A Multi-Omics and Technological Perspective

**DOI:** 10.3390/plants14060978

**Published:** 2025-03-20

**Authors:** Chunhua Zhang, Wenting Li, Yahan Wu, Shengli Li, Bao Hua, Haizhou Sun

**Affiliations:** 1Institute of Animal Nutrition and Feed, Inner Mongolia Academy of Agricultural & Animal Husbandry Sciences, Inner Mongolia, Hohhot 010031, China; zhangch-602@163.com (C.Z.); wuclwt@126.com (W.L.); wuya_han@126.com (Y.W.); lshl2000@163.com (S.L.); baohua009@126.com (B.H.); 2Key Laboratory of Grass-Feeding Livestock Healthy Breeding and Livestock Product Quality Control (Co-Construction by Ministry and Province), Ministry of Agriculture and Rural Affairs, Hohhot 010031, China; 3Inner Mongolia Key Laboratory of Herbivore Nutrition Science, Hohhot 010031, China

**Keywords:** chloroplast genome, growth–immunity trade-off, crop yield, sustainable agriculture, transcription, omics resources

## Abstract

Chloroplasts are important in plant growth, development, and defense mechanisms, making them central to addressing global agricultural challenges. This review explores the multi-faceted contributions of chloroplasts, including photosynthesis, hormone biosynthesis, and stress signaling, which orchestrate the trade-off between growth and defense. Advancements in chloroplast genomics, transcription, translation, and proteomics have deepened our understanding of their regulatory functions and interactions with nuclear-encoded proteins. Case studies have demonstrated the potential of chloroplast-targeted strategies, such as the expression of elongation factor EF-2 for heat tolerance and flavodiiron proteins for drought resilience, to enhance crop productivity and stress adaptation. Future research directions should focus on the need for integrating omics data with nanotechnology and synthetic biology to develop sustainable and resilient agricultural systems. This review uniquely integrates recent advancements in chloroplast genomics, transcriptional regulation, and synthetic biology to present a holistic perspective on optimizing plant growth and stress tolerance. We emphasize the role of chloroplast-driven trade-off in balancing growth and immunity, leveraging omics technologies and emerging biotechnological innovations. This comprehensive approach offers new insights into sustainable agricultural practices, making it a significant contribution to the field.

## 1. Introduction

Chloroplasts are essential organelles in plant cells and are primarily responsible for photosynthesis, which sustains plant metabolism by converting light energy into chemical energy [1]. Beyond photosynthesis, chloroplasts play a critical role in synthesizing biomolecules, including phytohormones and secondary metabolites, which regulate growth, development, and stress responses [2,3,4,5]. In an era of increasing global food demand and environmental challenges, understanding the role of chloroplasts in balancing plant growth and defense has become a crucial research focus.

Plants constantly encounter abiotic and biotic stresses, such as drought, light, temperature, pathogen attacks, and nutrient deficiencies [6]. To survive these challenges, they must carefully regulate resource allocation between growth and defense, a phenomenon known as the growth–defense trade-off [7]. This trade-off ensures that plants can adapt to changing environmental conditions by adjusting their metabolic priorities [8]. Understanding the growth–immunity trade-off is essential for advancing agriculture, ecology, and plant breeding [9]. Refining this balance can enhance crop productivity while reducing dependency on chemical inputs for pest and disease management. Recent studies have highlighted the regulatory roles of transcription factors in mediating these trade-offs between growth and immunity [9]. Chloroplasts serve as a central hub in this regulation by controlling photosynthetic efficiency, reactive oxygen species (ROS) signaling, and hormone biosynthesis, all of which influence how plants respond to stress [10,11]. Recent advancements in chloroplast genomics, transcriptomics, and synthetic biology have explored new insights into how chloroplasts mediate stress adaptations [12,13]. For instance, comparative analyses of the chloroplast genome have identified genetic variations that contribute to temperature stress adaptation in plants [14,15,16]. Additionally, studies have shown that chloroplast elongation factors can simultaneously promote yield and defense, effectively breaking the traditional growth–defense trade-off [17]. Furthermore, transcriptomics analyses have revealed that chloroplasts play a significant role in integrating signals during pathogen infection and plant defense responses [18].

Therefore, the engineering of chloroplasts is emerging as a significant potential means for controlling these trade-offs while enhancing crop yield [19,20,21,22,23]. Despite the extensive research on chloroplast functions in photosynthesis and plant metabolism, limited studies have focused on integrating advanced technologies that mediate the growth–defense trade-off at the chloroplast level. This review provides a novel perspective of chloroplast functions by systematically integrating recent findings with emerging technologies. Furthermore, it discusses the key regulatory functions performed by chloroplasts, such as hormone synthesis, stress signaling pathways, and transcription factor interactions, and advanced biotechnological techniques, such as chloroplast engineering and synthetic biology, to increase plant tolerance. By combining these studies, this review will provide insights into sustainable practices to improve the ability of crops to further withstand ongoing environmental changes.

## 2. Chloroplast Omics

### 2.1. Chloroplast Genome Structural Composition

The chloroplast genome (cpDNA) plays a crucial role in plant metabolism, encoding essential genes for photosynthesis, energy production, and other vital functions. The size and gene content of cpDNA exhibit considerable variation across plant species, with differences in genome size, structure, and the number of genes, reflecting the diversity in plant adaptation and evolution [24]. The first two plants in which the chloroplast genome (cpDNA) was successfully sequenced were tobacco and liverwort. Across plant species, the plastid gene content varies significantly, ranging from fully functional genomes (110–150 genes in photosynthetic plants) to highly reduced genomes in non-photosynthetic parasitic plants, which may retain as few as 20–50 genes essential for transcription and translation (e.g., *Cucuta* spp. and *Epifagus virginiana*) [25].

However, to date, no known plastid genome has been found to be completely devoid of genes. Even in the highly reduced plastid genome of non-photosynthetic plants, essential genes for transcription, translation, and protein import are retained. For instance, *Rafflesia lagascae* exhibits extreme plastome reduction, with a size of 11.5 kb, yet it retains a minimal set of genes, e.g., *atp*, *ndh*, *pet*, *psa*, *psb*, *rbcL*, rrn16, rrn23, rps7, rps11, rps16, and transfer RNA genes, as well as matK, accD, ycf2, and multiple nongenic regions from the inverted repeats for photosynthesis and energy production [26]. Similarly, the parasitic plant *Epifagus virginiana*, with a plastome size of 70,028 bp, has lost all photosynthetic genes but still contains 42 genes related to plastic maintenance [27]. These findings reinforce the fact that while plastid gene loss is extensive in some species, complete gene elimination has not been observed.

The chloroplast genome in photosynthetic plants typically ranges from approximately 120 kbp to 247 kbp in size [28]. However, a few plants possess extremely small chloroplast genomes, e.g., *Asarum minus*, which has 15 kbp, while a few others have very large chloroplast genomes, such as *Floydiella terrestris*, which possesses 521 kbp [29]. These genomes encode various functional gene classes, including protein-coding genes, tRNAs, and rRNAs, all contributing to chloroplast functions and plant growth. The number of genes within cpDNA is highly variable and influences a plant’s ability to adapt and thrive [30]. For instance, the cpDNA of *Arabidopsis thaliana* comprises 131 genes, including 97 protein-coding genes, 37 tRNA genes, and 4 rRNA genes, as well as two inverted repeats (IRs), 26,263 bp in size, each separated by a 84,197 bp large-single-copy (LSC) region and a 17,780 bp small-single-copy (SSC) region, supporting efficient photosynthesis and energy production [31]. The chloroplast genome sequences of 13 *Lamiaceae* species range from 149,081 to 152,312 bp and have two inverted repeats (IRs), with an average GC content of 27%, comprising 131–133 annotated genes, including 86–88 protein-coding genes, 27–28 tRNA genes, and 8 rRNA genes [32]. Monocots such as maize (*Zea mays*) have 140,387 bp and a pair of inverted repeats, IRa and IRb, which are separated by a small-single-copy (SSC; 12,536 bp) region and large-single-copy (LSC; 82,355 bp) region [33]. Legumes such as lima bean contain cpDNA with 150,902 bp, featuring a typical quadripartite structure that includes a pair of 26,543 bp inverted repeats (IRs), a 80,218 bp large-single-copy (LSC) region, and a 17,598 bp small-single-copy (SSC) region [34].

### 2.2. Plastid Genome Reduction in Parasitic and Non-Photosynthetic Plants

In contrast to photosynthetic plants, parasitic and non-photosynthetic plants exhibit significant reductions in both chloroplast number and gene content, due to the loss of photosynthetic-related genes [35]. For example, parasitic plants such as *Cuscuta* have fewer than 50 genes in their cpDNA, reflecting their reliance on the host plant. Conversely, species with more complex genomes such as *Zonal geranium* (217,942 bp), *Quercus ningangensis* (160,736 bp), and *Sicyos angulatus* (154,986 bp) maintain larger plastic genomes, often featuring well-defined inverted repeat (IR), large-single-copy (LSC), and small-single-copy (SSC) regions [36,37,38]. Similarly, the cpDNA of *Pistacia vera* (160,589 bp), *Datura* (1,555,497 bp), *Brugmansia* (1,555,497 bp), and *Sphaeropteris lepifera* (162,114 bp) follow the typical quadripartite structure found in most angiosperms [39,40,41], while *Xanthium spinosum* features a quadripartite circular structure with 152,422 bp [42]. Despite variations in their genome sizes, the IR regions of plastid genomes remain highly conserved across plant species, typically measuring 20,000 to 25,000 bp [43].

### 2.3. Chloroplast Proteomics and Functional Regulation

A proteomic analysis has identified between 2500 and 3500 distinct proteins within chloroplast, which far exceeds the coding capacity of the plastome itself [44]. Consequently, most plastid-localized proteins are encoded by nuclear genes; these proteins are synthesized in the cytosol as precursor proteins before being imported into the chloroplasts [45]. All major chloroplast protein complexes are subunits derived from nuclear and plastid genomes. However, coordinated expression between these two genetic systems is needed to ensure the proper assembly of protein complexes [4]. Recent bioanalytical and computational technology advancements have enabled the identification and quantification of thousands of proteins within complex mixtures. These technologies also help characterize post-translational modifications such as acetylation, glycosylation, and phosphorylation. Such insights into chloroplast proteomics can enhance our understanding of the intricate regulatory mechanism governing chloroplast functions in improving the growth and yield of crops (Table 1).

### 2.4. Chloroplast Transcription

During the early stages of chloroplast development, nuclear-encoded RNA polymerase (NEP) functionality is needed to transcribe genes such as rpoA, rpoB, rpoC1, and rpoC2, which encode components of the plastid-encoded RNA polymerase (PEP). Once synthesized, PEP becomes the primary enzyme responsible for transcribing photosynthetic genes, allowing for transition from early chloroplast development to a fully functional chloroplast [53]. In angiosperms, the transcription of chloroplast genes is mediated by two distinct RNA polymerases, PEP and NEP [53], where PEP is composed of multiple subunits, with the core subunits encoded by the chloroplast DNA (cpDNA), including the α (38 kDa) and β (120 kDa), β′ (85 kDa), and β″ (185 kDa) subunits [25]. Although most genes encoding PEP subunits have been translocated to the nuclear genome, essential genes for the core subunits remain in the cpDNA [4]. Conversely, NEP is characterized as a single-subunit enzyme that executes the entire transcription process autonomously, from promoter recognition to transcription termination, regardless of the structural characteristics of the DNA template [54]. NEP evolved from the duplication of the nuclear gene that encodes mitochondrial RNA polymerase and exhibits significant similarity to phage-type RNA polymerases, which are also composed of a single subunit. In angiosperms, three variants of NEP polymerases are recognized: RPOTp, which is present in monocots; RPOTp and RPOTmp, found in dicots; and RPOTm, which is localized in mitochondria. PEP and NEP are crucial for transcribing chloroplast genes [55,56]. Despite their distinct promoter recognition capabilities, many chloroplast genes have promoters that can be recognized by both PEP and NEP, indicating a collaborative role in gene expression within chloroplasts. Plastid mRNA processing is a critical regulatory step in chloroplast gene expression [53]. Before functional chloroplast transcripts can be generated, their large polycistronic precursors must undergo extensive post-transcriptional modifications [57]. These processing events include endonucleolytic cleavage by RNases, exonucleolytic trimming, RNA splicing, and RNA stability control. Endonucleolytic cleavage, primarily carried out by ribonucleases such as RNase J and RNase E-like enzymes, generates mature monocistronic or oligocistronic transcripts from primary polycistronic transcripts [57], which is essential for proper gene regulation and efficient translation.

Plastid transcript maturation involves multiple post-transcriptional modifications to ensure proper gene expression and functionality. Exonucleolytic trimming is crucial in refining transcript ends, enhancing their stability, and optimizing translation efficiency [58]. Another key step in plastid transcript processing is RNA splicing, which involves group I and II introns. This process is facilitated by various RNA-binding proteins and maturases, which are responsible for removing introns to generate functional mRNA [59]. Additionally, nucleoid-associated proteins play an essential role in plastid transcript maturation and stability. Another study determined that nucleoid proteins, such as pTAC components, are required for the maturation and stability of plastid transcripts. The loss of these proteins has been linked to severe developmental defects, including albino or chlorotic phenotypes, indicating their essential function in chloroplast biogenesis [59]. This interplay is vital for maintaining chloroplast functionality, which supports metabolic efficiency, stress adaptation, and overall plant health. While transcriptional regulation influences photosynthesis and resilience to environmental stress, its direct impact on agricultural productivity depends on multiple genetic and environmental factors.

### 2.5. Chloroplast Translation

Chloroplasts have a semi-autonomous expression apparatus, reflecting their evolutionary origin from cyanobacteria. The final stage of gene expression translation occurs within the chloroplast stroma, helped by bacterial-type 70S ribosome and a variety of translational factors encoded by both the nuclear and chloroplast genomes [60]. Biochemical techniques for protein identification have been pivotal in characterizing the components of the plastid translation machinery, revealing chloroplast ribosomal proteins derived from the nuclear genome, as well as translation initiation and elongation factors, and tRNA synthetase [61]. The role of translational factors in regulating chloroplast gene expression is linked to plant ontogeny. Research during the initial stages of chloroplast differentiation has demonstrated elevated activity of PEP polymerase, leading to a significant increase in mRNA level coding for protein integral to the photosynthetic apparatus [62]. However, the mechanism governing translation regulation during this phase remains poorly understood. A notable translation regulator in *Arabidopsis thaliana* is the nuclear-encoded ATAB2 factor, which ensures proper alignment of polycistronic mRNA from the PSAA-PSAB and PSBC-PSBD operons on the ribosomes [63]. Another critical nuclear-encoded protein is the pentatricopeptide repeat protein (LPE1). Its absence results in diminished translation of psbN and a complete loss of psbJ translation [64]. Nuclear genes that encode the chloroplast ribosome-associated (CRASS) protein have recently been identified. CRASS interacts with the 30s ribosomal subunit components directly or through intermediary factors [65]. While not essential for plant survival under optimal growth conditions, CRASS enhances ribosomal functions under stress, particularly during cold exposure, by acting as a regulatory or stabilizing agent for the 30S subunit [66]. The major proteins responsible for chloroplast protein import include the Toc-Tic translocon system, HSP50, and HSP93, a member of the HSP100 protein family, which play crucial roles in motor protein import [67]. The primary function of these proteins is to convert the intermediate products into final proteins and translocate them to their final destination via various pathways [67]. Several envelope proteins are present in plastids’ inner and outer membranes, with many of their functions identified, while few remain unknown. The outer membrane envelope contains proteins such as LACS9, MGD1, and SFR2v (responsible for lipid biosynthesis); HXK1 and NPL1 (responsible for signaling and response); and Emb1211, Emb2004, Emb2458, and Emb2737 (responsible for embryo development) [68]. Additionally, TOC34, TOC64-III, TOC75-III, and TOC75-V facilitate general protein import, while LPTD and OEP16-1 are responsible for transport [69]. Similarly, IEP37, DIT1, DIT2.1, KEA1, and MEX1 (transport proteins); CJD1, TIC22-IV, TIC40, and TIC32-IVb (general import proteins); ATP5, PCAP1, and SAMC1 (metabolic proteins); LOX1, AVP-3, ESM1, and PYK10 (signaling and response proteins); and FAD8, HPL1, SQD2, and SUR2 (lipid biosynthesis proteins) are envelop proteins on the inner stroma of the plastid [70]. Furthermore, the CHLORAD (Chloroplast-Associated Protein Degradation System) has been identified as a key ubiquitin-mediated degradation pathway in chloroplasts [71]. The CHLORAD functions in chloroplast protein homeostasis by regulating the degradation of TOC translocon components through ubiquitination, retrotranslocation, and proteasomal degradation [71]. This system, involving SP1, SP2, and CDC48, plays a crucial role in remodeling the chloroplast proteome in response to developmental and environmental signals [72]. Understanding these regulatory roles can open avenues for genetic strategies to boost crop productivity and resistance, aligning with the broader goal of enhancing agricultural productivity under varying conditions (Figure 1).

## 3. Chloroplast Databases and Tools: Resources for Genomic and Functional Insight

### 3.1. Database Resources

The chloroplast genome database (ChloroplastDB) is an essential Web-based resource for researchers, designed specifically to host fully sequenced plastid genomes by providing comprehensive comparative genomic studies. It provides crucial insight into the evolution of plastids across diverse plant species and supports phylogenetic analyses [73]. ChloroplastDB includes the genome of 3832 plant species, spanning 1527 genera across 256 families [74]. This database is regularly updated with a new sequence as they become available on NCBI (https://www.ncbi.nlm.nih.gov/ accessed on 15 December 2024). Currently, the database contains 495,769 gene records and 1,245,379 feature records, serving as a comprehensive resource for chloroplast genome research [75].

The Chloroplast Genome Information Resource (CGIR) (https://ngdc.cncb.ac.cn/cgir/ accessed on 15 December 2024) is a comprehensive and curated database designed to support in-depth analyses of chloroplast genomes [76]. It integrates and standardizes data on chloroplast genes, simple sequence repeats (SSRs), and DNA signature sequences (DSSs), making it an invaluable tool for molecular and evolutionary biology research. CGIR contains 19,388 chloroplast genome assemblies from 11,946 plant species, along with detailed gene annotation, SSRs, and DSSs [77]. Notably, CGIR features 1170 newly sequenced chloroplast genomes from 718 species, with genomes from 307 species being reported for the first time, including one new family (Juncaginaceae) and 53 genera [78]. This extensive database is crucial for genomic comparisons, evolutionary studies, and the identification of unique chloroplast markers, supporting research on plant genetic diversity and phylogenetic relationships.

Chloroplast Genome View (CPGView) is a versatile tool designed for visualizing chloroplast genomes. It generates three distinct maps: (i) the distribution of genes, variable sites, and repetitive sequences; (ii) the adjusted exon–intron structure of cis-splicing genes using a coordinate scaling algorithm; and (iii) the structure of trans-splicing gene *rps12* [79]. To validate its accuracy, 31 chloroplast genomes from 31 genera across 22 families were sequenced, assembled, and annotated, with CPGView accurately drawing maps for all [79]. Additionally, the tool was tested on 5998 publicly available chloroplast genomes from 2513 genera spanning 533 families. It successfully plotted maps for 5882 genomes and only failed for 116, which were found to have annotation errors requiring manual correction [80]. These tests highlight CPGView’s utility in identifying annotation issues and accurately analyzing chloroplast genome structures.

CHLOROBOX (https://chlorobox.mpimp-golm.mpg.de/ accessed on 15 December 2024) and Plastid Genome Resources (Plastome) work together to advance chloroplast genomic research by addressing different yet interconnected aspects of the field [80]. CHLOROBOX primarily facilitates the conversion of revised GenBank files, generated by GenSeq into formats such as Sequin or Banklt, which are required for NCBI/EMBL/DDBJ database submission [81]. On the other hand, Plastome serves as a comprehensive repository of plastid genomes from a wide array of taxa, enabling in-depth structural and evolutionary studies. Together, these resources bridge the gap between functional genomics and comparative analyses [82]. By integrating promoter predictions from CHLOROBOX with genome-wide structural data provided by Plastome, researchers can gain a more comprehensive understanding of plastid genome organization, gene regulation, and evolutionary dynamics across diverse plant species.

### 3.2. Chloroplast Research Databases

Chloroplast research benefits from diverse databases that provide essential resources for genomic, functional, and evolutionary studies. NCBI GenBank (https://www.ncbi.nlm.nih.gov/genbank/ accessed on 15 December 2024) serves as the central repository for chloroplast genome sequences from various species, enabling genomic research and annotation efforts [83]. Ensembl Plant (https://plants.ensembl.org/index.html accessed on 15 December 2024) offers a robust platform with annotated plant genomes, including chloroplast-specific sequences, facilitating comparative analyses and explorations of gene function [84]. UniProt (https://www.uniprot.org/ accessed on 15 December 2024) provides a curated database of protein sequences, emphasizing chloroplast-specific proteins, to support proteomic studies in photosynthesis and other critical biological pathways [85]. KEGG (Kyoto Encyclopedia Genes and Genomes) (https://www.genome.jp/kegg/ accessed on 15 December 2024) provides map pathways and metabolic processes related to chloroplast functions, such as the photosynthesis and biosynthesis of secondary metabolites [86]. Phytozome (https://phytozome-next.jgi.doe.gov/ accessed on 15 December 2024) enables evolutionary and comparative analyses by providing chloroplast genome data from diverse plant species [87]. The Plant Genome Database (PGDB) (https://pgdbj.jp/sp_en/index accessed on 15 December 2024) focuses on phylogenetic studies and cross-species genome comparisons, enriching evolutionary research [88]. Collectively, these databases form an interconnected ecosystem that advances the understanding of chloroplast genomics, functionality, and evolutionary dynamics (Figure 2).

## 4. Major Functions Performed by Chloroplast

### 4.1. Chlorophyll Synthesis 

Chlorophyll synthesis in chloroplast begins with activating the amino acid glutamate, which is converted into glutamyl-tRNA and forms the preliminary structure for the synthesis pathway [89]. Notably, glutamyl-tRNA plays a crucial role in the synthesis of glutamate-1-semialdehyde, which is subsequently transformed into 5-aminolevulinic acid (ALA) by glutamate-1-semialdehyde aminotransferase. ALA dehydratase synthesizes and condenses multiple ALA molecules to produce porphobilinogen (PBG), a monopyrrole structure [90]. In the next step, the porphobilinogen deaminase enzyme links the four PBG molecules to form a tetra pyrrole that is cyclized by uroporphyrinogen III synthase to create uroporphyrinogen III, an essential intermediate [91]. This molecule is then decarboxylated by uroporphyrinogen decarboxylase to form coproporphyrinogen III, which is oxidized by coproporphyrinogen oxidase and further by protoporphyrinogen oxidase to yield protoporphyrin IX [92]. Then, the enzyme Mg-chelatase catalyzes the insertion of Mg^2+^ into protoporphyrin IX, distinguishing chlorophyll synthesis from heme biosynthesis. Genome uncoupler 4 (GUN4) plays a pivotal role in this process by binding protoporphyrin IX and the product of Mg-chelatase by enhancing the activity of the enzyme to produce Mg-protoporphyrin IX, which is then converted into protochlorophyllide by a cyclase enzyme [93]. The reduction of protochlorophyllide to chlorophyllide mediated by protochlorophyllide reductase is a light-dependent reaction and a pivotal point in chlorophyll synthesis regulation [94]. Finally, chlorophyll synthase attaches a phytol tail to chlorophyllide to produce chlorophyll b. In contrast, chlorophyllide, an oxygenase, converts some chlorophyll into chlorophyll b to perform various light absorption reactions [95]. Chlorophyll synthesis within chloroplast provides valuable insight into optimizing photosynthetic efficiency, which holds significant potential for enhancing crop yield and bolstering plant defense mechanisms.

### 4.2. Photosynthesis 

Chloroplasts involve several key processes, including light capture, electron transport, and ATP generation. Initially, light energy is absorbed by chlorophyll molecules in photosystem two (PSII), which excites electrons and initiates the electron transport chain [96]. The excited electrons are passed to plastoquinone, then through the cytochrome b_6_f complex, and finally to plastocyanin before reaching photosystem one (PSI). The electron transport between PSII and PSI helps establish a proton gradient across the thylakoid membrane, driving ATP synthesis through ATP synthase [97]. Upon reaching PSI, electrons are further energized via light absorption, then transferred to ferredoxin. Here, cyclic electron flow (CEF) plays a critical role in photosynthesis [98]. The antimycin A-sensitive CEF (AA-sensitive CEF) was discovered by Arnon and colleagues over fifty years ago, with its mechanism only recently elucidated [99]. This pathway recycles electrons from ferredoxin back to plastoquinone, allowing for the generation of additional ATP without the production of NADPH. The proteins PGR5 and PGR1 in thylakoid membranes are implicated in this CEF process [100]. Although their precise function in CEF in vivo remains contentious due to technical challenges, recent biochemical evidence confirms that PGR1–PGR5 complexes exhibit ferredoxin-plastoquinone reductase (FQR) activity in vitro [15]. These complexes seem to shuttle electrons between PSI and the cytochrome B6F complex in flowering plants. In contrast, in the green alga Chlamydomonas, PGR1 but not PGR5 has been observed within a PSI–cytochrome B6F supercomplex that inherently supports CEF [101]. The NDH complex represents another significant pathway mediating CEF. Unlike the PGR1–PGR5 complex, the NDH complex, which resembles NADH–hydrogenase complexes in bacteria and mitochondria, accepts electrons from ferredoxin rather than NADPH [102]. Recent findings by the Shikanai group identified new subunits of the plant NDH complex, CRR31, J, and L, with CRR31 providing a docking site for ferredoxin [103]. This discovery led researchers to propose that the term NDH be understood as an NADH–hydrogenase-like complex. The NDH complex functions as an FQR, facilitating the movement of electrons and enhancing ATP production [104]. Structurally, the NDH complex has been shown to form supercomplexes with PSI, as demonstrated through genetic, biochemical, and electron microscopy studies. These supercomplexes typically involve two PSI units binding to one NDH complex, with the light-harvesting complex proteins LHCA5 and LHCA6 acting as connectors [105]. Intriguingly, the formation of PSI–NDH supercomplexes coupled with light-harvesting proteins predates the evolution of vascular plants, as seen in the moss *Physcomitrella patens* [106]. This evolutionary insight highlights the integral role of CEF and NDH in maintaining photosynthetic efficiency across plant lineages. The central role of chloroplast in photosynthetic efficiency and energy production provides a strong foundation for strategies aimed at improving crop yield and enhancing plant defense mechanisms against environmental stressors.

### 4.3. ABA Biosynthesis

ABA (abscisic acid) is a 15-carbon compound synthesized through the formation of 40-carbon intermediates. The initial stages of ABA biosynthesis, involving the interconversion of various 40-carbon intermediates, take place in the chloroplast [107]. Notably, the conversion from xanthophylls requires NADPH and O_2_. 9’-cis-neoxanthin and 9-cis-violaxanthin are subsequently cleaved by 9-cis-epoxycarotenoid dioxygenase (NCED) within the chloroplast to produce the 15-carbon compound xanthoxin [108]. Xanthoxin is then transported from the chloroplast to the cytosol, where xanthoxin dehydrogenase converts it into abscisic aldehyde [109]. The final oxidation step, in which abscisic aldehyde is transformed into ABA, is catalyzed by ABA aldehyde oxidase and requires the presence of O_2_. ABA may undergo modification, such as glycosylation or esterification, leading to an inactive state [14]. Under stress conditions, ABA is released from its conjugated form, ABA-glucose, due to the action of β-glucosidase. ABA functions as a critical regulator of plant growth and development and biotic and abiotic stress responses [110]. It plays a pivotal role in inducing stomatal closure under drought and high-salinity conditions, reducing gas exchange essential for photosynthetic carbon fixation. Moreover, ABA treatment has been shown to elevate the expression of OB1G, a tomato homolog of the ABA-responsive gene, and to increase H_2_O_2_ levels in both the apoplast and chloroplast [111]. At the whole-plant scale, it is essential to recognize that while ABA facilitates ROS production in non-seed tissues, it inhibits ROS generation in seeds. Additionally, ABA treatment induces nitric oxide production, mediated by nitrate reductase and a plasma membrane-bound enzyme (Purified Plasma Membranes (PMs)). ABA biosynthesis and signaling, and their elongation factors (internal and external signals) highlight their influence on stress adaptation and growth regulation, offering promising avenues to enhance crop yield and resistance in challenging environments.

### 4.4. Ethylene Hormone

ET (ethylene) is a two-carbon gaseous plant hormone that plays a significant role in plant growth, development, and stress responses [112]. While ET itself is not synthesized within the chloroplast, its precursor, methionine (met), is produced there. ET biosynthesis involves a three-step process. The conversion of methylenocycloprotein one carboxylic acid (ACC) is facilitated by ACC synthase, and then the conversion of ACC to ET by ACC oxidase [5]. *Arabidopsis* contains three cobalt and independent methylene synthases (MSs), with MS three localized in the chloroplast, essential for generating methyl de novo synthesis homocysteine within the chloroplast [113]. MS one and MS two are cytosolic and likely contribute to methyl generation during the activated metal cycle. ET regulates various physiological processes, including modulating stomatal aperture, which can affect photosynthesis [114]. The impact of ET on stomatal movements has shown variability across different studies. Research using epidermal peels has indicated that ET can influence stomatal opening in diverse plant species [115]. Additionally, ET functions as a crucial mediator in plant responses to pathogens. It mitigates symptom development caused by necrotrophic pathogens while promoting cell death in interactions with biotrophic and hemibiotrophic pathogens [116]. For instance, *Arabidopsis* mutants with reduced ET sensitivity have demonstrated increased susceptibility to the necrotrophic fungus but increased resistance to the hemibiotrophic pathogen *Pseudomonas syringae* pv. Tomato (Pst) and the biotrophic pathogen *Xanthomonas campestris* pv. Campestris [117]. Similarly, ET-insensitive soybean mutants exhibited more pronounced symptoms when the necrotrophic fungi *Septoria glycines* and *Rhizoctonia solani* were infected. Still, they displayed reduced symptom severity when infected with hemibiotrophic *Pseudomonas syringae* pv. glycinea and *Phytophthora sojae* [118]. The interplay between chloroplast metabolites and ethylene biosynthesis offers additional opportunities to enhance crop yield and fortify plant defense against diverse environmental challenges.

### 4.5. Jasmonic Acid Synthesis

Jasmonic acid (JA) is a twelve-carbon oxygenated fatty acid derivative synthesized within chloroplasts and peroxisomes [119]. Its biosynthesis begins in the chloroplast, where linolenic acid (18:3), an 18-carbon polyunsaturated fatty acid, is liberated from membrane lipids and subsequently oxidized at the C13 position by lipoxygenase, with O_2_ serving as a substrate [120]. This oxidation results in the formation of an intermediate, which is further cyclized into 12-oxophytodienoic acid (OPDA) through the coordinated actions of allene oxide synthase and allene oxide cyclase. OPDA is then transported to the peroxisome, where it undergoes reduction to form 3-oxo-2-(2’-pentenyl) cyclopentane-1-octanoic acid (OPC8), catalyzed by OPDA reductase, utilizing NADPH as a reducing agent [121]. Subsequent beta-oxidation processes, which require ATP and O_2_, convert OPC8 into JA. JA can undergo various modifications post synthesis, such as amino acid conjugation, methylation, sulfation, glycosylation, and hydroxylation, facilitating precise regulation of its accumulation, bioactivity, and mobility [122]. JA is a pivotal regulator of plant development and responds to abiotic and biotic stresses. The application of exogenous JA at a concentration of 1 μM has been shown to significantly reduce photosynthetic pigment levels in potato leaves [123]. Additionally, methyl jasmonate (MeJA) treatment impairs photosynthetic electron transport and carbon fixation in *Arabidopsis* protoplasts and in *Vicia faba* and rice seedlings [123]. As demonstrated in barley and citrus seedlings, JA application has also been linked to increased abscisic acid (ABA) levels. This suggests a role for JA in facilitating ABA accumulation during drought stress, consistent with the synergistic relationship between JA and ABA in activating drought-responsive pathways [124]. Specifically, JA and ABA cooperatively enhance the MYC branch of the JA signaling pathway, which induces the expression of wounding-responsive genes while suppressing pathogen-responsive genes [125]. Conversely, JA acts in conjunction with ethylene (ET) in the ERF branch of the signaling pathway to promote pathogen-responsive genes and to downregulate wounding-responsive genes [126]. The response to MeJA can vary depending on the plant species and developmental stage, influencing ET production in seedlings, fruits, and seeds either positively or negatively. This interaction of JA with other hormonal pathways highlights chloroplast’s role in enhancing crop productivity and resistance against pathogen-induced challenges.

### 4.6. Synthesis of Salicylic Acid

Chloroplast synthesizes SA (salicylic acid), a seven-carbon phenolic compound, through two main pathways: the isochorismate pathway within the chloroplast and the phenylalanine ammonia lyase (PAL) pathway [127]. In the chloroplast, chorismate is converted to isochorismate by the enzyme isochorismate synthase (ICS). Isochorismate is subsequently converted into SA and pyruvate through the action of isochorismate pyruvate lyase. Although the gene encoding this enzyme in plants has yet to be identified [128], in *Arabidopsis thaliana*, two ICS genes, ICS1 and ICS2, are known. A genetic analysis revealed that under pathogenic attacks or stress conditions, ICS1 mutants accumulate approximately 5–10% of the SA observed in wild-type plants, and ics1 and ics2 mutants accumulate around 4% of the wild-type SA [129]. These findings indicate that the isochorismate pathway is the predominant route for SA biosynthesis in plants. Additionally, within the chloroplast, chorismate is converted into phenylalanine, which is subsequently exported to the cytosol [130]. Phenylalanine is converted into cinnamate through the catalytic activity of phenylalanine ammonia-lyase (PAL), and cinnamate is further transformed into SA via beta-oxidation or non-oxidative routes [127]. Once synthesized, SA undergoes modifications such as glycosylation, methylation, amino acid conjugation, and hydroxylation, facilitating precise regulation of its accumulation, biological activity, and mobility [131]. SA functions as a regulator of plant growth and development and a mediator of responses to both abiotic and biotic stress conditions. However, the impact of SA on photosynthesis under optimal growth conditions has been debated [132]. For instance, the foliar application of SA solutions on soybean shoots significantly enhanced plant growth without altering the rate of photosynthesis.

Studies have shown that applying SA at an optimal concentration (5–10 mM) on crops such as Indian mustard, maize, and soybean markedly improved their photosynthetic rates. In contrast, higher concentrations exhibited inhibitory effects [133]. Under stress conditions, pre-treatment with SA has been observed to mitigate stress-induced damage, thereby enhancing photosynthetic efficiency [134]. These observations underscore that the influence of SA on photosynthesis varies based on the plant species, the method and duration of application, and the growth conditions.

### 4.7. ROS Production

In chloroplasts, PSI (photosystem I) and PSII (photosystem II) serve as primary sources of reactive oxygen species (ROS) generation. PSI acts as a primary producer of superoxide (O_2_-) during photosynthesis [135]. Oxygen is continuously reduced to O_2_- by PSI, which is subsequently converted into hydrogen peroxide (H_2_O_2_) and molecular oxygen (O_2_) by Cu/Zn superoxide dismutase (Cu/ZnSOD) associated with PSI [136]. PSII predominantly generates singlet oxygen (1O_2_) in the photosynthetic process. Ground-state oxygen (3O_2_) is regularly excited to 1O_2_ by triplet-excited state chlorophyll (3P680) within the PSII reaction center [137]. ROS production in chloroplasts has beneficial and detrimental effects on photosynthesis, similar to ROSs generated in other cellular compartments. Chloroplastic ROSs play an essential role in the hypersensitive response in plants [138]. For instance, enhanced ROS accumulation was observed to precede localized cell death in wild-type tobacco leaves infiltrated with the non-host pathogen Xcv. However, tobacco plants with impaired chloroplastic ROS production displayed a significant reduction in localized cell death following Xcv inoculation [139]. These plants expressed cyanobacterial flavodoxin, which inhibits chloroplastic ROS production during pathogen challenge. The dual role of chloroplast-generated ROSs in stress signaling underscores their pivotal role in enhancing crop yield and resistance against pathogenic stress (Figure 3).

## 5. The Role of Chloroplasts in the Trade-Off Between Plant Defense and Growth

### 5.1. The Role of Chloroplasts in Trade-Off 

Chloroplast plays a pivotal dual role in energy production and in the generation of defense-related signaling molecules or their precursors [2]. Within this dual functionality, the trade-off between growth and defense mechanisms in plants is highlighted. Chloroplasts facilitate interactions between photosynthesis and defense signaling due to their concurrent roles. Photosynthesis produces essential carbon skeletons and energy and reduces the power needed to synthesize defense-related molecules [140]. These molecules include the early steps of ABA (abscisic acid), ET (ethylene), and JA (jasmonic acid) biosynthesis, which occur within the chloroplast. Additionally, calcium (Ca^2+^) is generated by thylakoid membrane-localized CGS. These defense-related molecules can reciprocally impact chloroplast functions by influencing the regulation of photosynthetic genes [5]. This regulation affects chloroplast-encoded and nuclear-encoded photosynthetic protein genes, demonstrating how defense responses can influence photosynthetic efficiency and plant growth [141]. Pathogens target chloroplasts to disrupt their function, aiding in their infection strategy. Some pathogen effectors (e.g., *Fg03600*, a chloroplast-targeting effector) utilize cleavable chloroplast transit peptides (e.g., RVRPS4 and HAPK1) to translocate into chloroplasts, while others (e.g., HAP1 and HAPN1) employ non-cleavable transit peptides [142]. Once inside the chloroplast, effectors can interact with various chloroplastic proteins such as PTF1, CBSX2, HSP70, and PSBQ. They may manipulate the chloroplast structure and function, affecting thylakoid remodeling, photosynthetic gene expression, photosynthetic water splitting, electron transport, enzyme redox status, and auxin biosynthesis [1]. In addition to direct effector activity, pathogens produce phytotoxins that disrupt chloroplast function to suppress host defenses and enhance infection. Toxins and phaseolotoxins specifically inhibit the activities of chloroplastic enzymes, glutamine synthetase, and ornithine carbamoyl transferase, respectively [143]. The phytotoxin coronatine (COR) modulates JA signaling, leading to chlorophyll degradation and the suppression of auxin biosynthesis, ultimately contributing to chlorosis in plant tissues [144]. This balance highlights the chloroplast’s trade-off between supporting photosynthesis or growth and engaging in defense responses. Pathogens’ manipulation of chloroplast functions and the synthesis of signaling molecules within organelles underscore its complex role at the intersection of plant growth, pathogen attack, and defense signaling.

### 5.2. Success Stories of Trade-Off Applications in Agriculture

Chloroplasts host light-driven electron transport reactions that produce energized electrons necessary for many metabolic processes [145]. However, imbalances in electron flow, which can arise under variable light conditions or when electron sinks are restricted (such as during stomatal closure), pose significant risks due to the generation of highly reactive singlet oxygen (^1^O_2_) in photosystem II (PSII) or superoxide (O^2−^), ultimately leading to hydrogen peroxide (H_2_O_2_) formation in photosystem I (PSI) [146]. The accumulation of these reactive oxygen species (ROSs) can be damaging, necessitating finely tuned cellular responses to match environmental conditions through chloroplast-generated signaling [147]. One critical regulatory mechanism involves the redox poise of plastoquinone, influenced by the excitation balance between the photosystems. This redox state activates proteins such as STN7 and CSK (an SRC family kinase), which can modulate plastid gene expression to adjust the stoichiometry of the photosystems, thereby maintaining homeostasis [146]. These redox changes also trigger extensive modifications in nuclear gene expression and overall cellular metabolism. Under high light exposure, ^1^O_2_ acts as a potent signal initiating the expression of stress-responsive genes, often encoding antioxidant proteins or triggering programmed cell death response [148]. The absence of plastid-localized executor1 (EX1) and executor2 (EX2) prevents both the gene expression responses and cell death associated with ^1^O_2_ accumulation, suggesting an essential role for these proteins in mediating ^1^O_2_ signaling, potentially through plastid transcriptional regulation [149]. H_2_O_2_ generated by chloroplasts under high light stress induces the transcription of antioxidant enzyme-encoding genes, such as APX2, and other stress-related genes locally and systemically by synthesizing the stress hormone abscisic acid (ABA) [150]. Recent research has identified three critical metabolites in plastid-to-nucleus signaling: 1. phosphoadenosine 5’-phosphate (PAP); 2. β-cyclocitrol; and 3. methylerythritol cyclodiphosphate (MECP). β-Cyclocitrol, an oxidation product of β-carotene induced by ^1^O_2_, mediates substantial transcriptional responses in the nucleus [151]. Due to the short-lived nature of ^1^O_2_ and its confinement within chloroplasts, the similarity between transcriptional responses to ^1^O_2_ and β-cyclotron suggests that β-cyclotron functions as a chloroplast-to-nucleus signal [152]. PAP accumulates in response to high light and drought stress, with its levels regulated by the chloroplast enzyme CEL1. MECP, a substrate of a rate-limiting enzyme in the MEP (methylerythritol phosphate) pathway, has been shown to act as a global stress sensor [153]. Mutants deficient in this enzyme exhibit constitutive expression of nuclear stress genes, heightened responses to abiotic stress, and resistance to biotrophic pathogens. Elevated MECP levels during stress in wild-type plants reinforce the idea that the MEP pathway contributes significantly to stress signaling [154]. Demonstrating the role of chloroplasts in regulating not only their functions but also those of the entire organism, chloroplast differentiation and the expression of photosynthetic genes are closely linked with the developmental transition of mesophyll cells from proliferation to expansion, as observed in *Arabidopsis thaliana* [155]. While more gradual in maize, this transition suggests an intricate developmental link between chloroplasts and mesophyll cells [156]. During senescence or in response to pathogen attacks, the chlorophyll catabolite pheophorbide can trigger cell death independently of light exposure, pointing to non-photo-oxidated pathways of programmed cell death that resemble mitochondrial involvement in mammalian apoptosis [157]. Understanding the complex interplay between plastid biogenesis, genetic regulation, protein import, division machinery, and nuclear gene expression within cellular differentiation remains essential [158]. Such integration is evident during developmental stages, as highlighted in maize leaf development, where plastid biogenesis coincides with the expression of light-responsive transcription factors and plastid-to-nucleus communication pathways [159]. Understanding this two-way relationship between chloroplasts and cellular differentiation is pivotal to the field yet remains in need of further exploration (Figure 3).

## 6. Chloroplast’s Role in Plant Growth and Development

### 6.1. Role of Elongation Factor in Heat Stress Tolerance

Plant heat stress triggers the synthesis of essential proteins, including elongation factors that play a crucial role in maintaining translational fidelity under thermal stress. While eukaryotic elongation factor EF-2 operates in the cytoplasm, chloroplasts process their own distinct set of elongation factors, including elongation factor Tu (EF-Tu), EF-G, EF-Ts, and EF-P, which are essential for plastid ribosomal functions and protein synthesis. Among these factors, chloroplast EF-Tu protects the photosynthetic apparatus from thermal damage [17].

The chloroplast EF-Tu is a highly conserved GTP-binding protein involved in the elongation phase of translation by facilitating aminoacyl-tRNA delivery to ribosomes [140]. Beyond its primary role in protein synthesis, EF-Tu acts as a molecular chaperone, preventing the thermal aggregation of newly synthesized proteins and stabilizing the chloroplast translation machinery under heat stress [160]. In maize and wheat, the increased accumulation of EF-Tu under heat stress has been linked to enhanced chloroplast stability and improved photosynthetic efficiency in heat-tolerant cultivars [161]. Such results have been observed in the heat-tolerant maize line ZPBL1304, which shows significant accumulation of EF-2 under thermal stress, whereas the heat-sensitive line ZPL389 shows no such induction [162]. In addition to chloroplast EF-Tu, other chloroplast elongation factors, such as EF-G and EF-Ts, contribute to translational efficiency under stress conditions [163]. EF-G facilitates ribosomal translocation and is involved in stress adaptation by ensuring proper ribosome movement and elongation cycle progression. EF-Ts, a guanine nucleotide exchange factor, assists in EF-Tu recycling, thereby sustaining the efficiency of protein synthesis under prolonged stress conditions [163]. The functional interplay among these elongation factors is crucial for maintaining chloroplast proteostasis, particularly in plants exposed to repeated or prolonged heat stress.

### 6.2. Regulation of Carbon Metabolism via Phosphoglucose Isomerase (PGI)

Phosphoglucose isomerase (PGI) facilitates the interconversion of fructose-6-phosphate (F6P) and glucose-6-phosphate (G6P), playing a crucial role in the biosynthesis of starch and sucrose [164]. Both plastidic and cytosolic isoforms are present in plant leaves. Using recombinant enzymes and isolated chloroplasts, isoforms of these PGIs were characterized. The findings reveal that *Arabidopsis thaliana* plastidic PGI exhibits a Michaelis constant (KM) for G6P that is three times higher than that for F6P, with erythrose-4-phosphate functioning as a significant regulatory molecule influencing PGI activity [165]. Additionally, the KM of spinach plastidic PGI was subjected to dynamic regulation, showing a 200% increase in dark compared to light conditions [166]. Furthermore, the heterologous expression of an *Arabidopsi*s cytosolic PGI targeting the plastids in *Nicotiana tabacum* impeded both starch synthesis and degradation [167]. These observations, along with the finding that plastidic PGI operates out of equilibrium, suggest that PGI plays a pivotal regulatory role, acting as a unidirectional control point to restrict G6P backflow into the Calvin–Benson cycle [168]. Therefore, the strategic manipulation of PGI could be leveraged to enhance carbon flux towards specific biotechnological targets. This presents exciting opportunities for metabolic engineering to improve plant productivity and optimize biotechnological processes.

### 6.3. Enhancing Drought Tolerance with Flavodiiron (FLV) Proteins

Plants have developed various mechanisms to tolerate short periods of drought stress, such as synthesizing essential metabolites for plant function via chloroplasts. In photosynthetic organisms, FLVs help to protect against photoinhibition by reducing oxygen at the non-heme diiron active site within their metallolactamase-like domain. FLV genes, which include Flavodiiron proteins, have been demonstrated to enhance stress tolerance in cyanobacteria [169]. To bolster drought resilience in plants, the FLV1 and FLV3 genes from *Synechocystis* were heterologously expressed in *Hordeum vulgare* (barley), with their protein products targeted for translocation to the chloroplasts [170]. Transgenic barley lines expressing FLV1 and FLV3 exhibited accelerated flowering, increased biomass accumulation, enhanced spike and grain production, and elevated total grain yield under drought conditions [170]. This improved growth performance was linked to the increased availability of soluble carbohydrates, an elevated turnover rate of amino acids, and reduced proline accumulation in the leaves [171]. The presence of FLV1 and FLV3 facilitated the maintenance of energy homeostasis in drought-stressed leaves by catalyzing the conversion of sucrose into glucose and fructose, which serve as immediate substrates for energy metabolism to support growth under water-deficient conditions. Incorporating FLV genes into the plant genome could enhance crop productivity in regions where drought stress poses a significant agricultural challenge [21].

### 6.4. Carotenoid Biosynthesis and the Role of DCLCYB1

β-carotene is an essential molecule for plant functioning and enhancing food quality in crops. Due to its significance, genetic engineering techniques have been employed to boost β-carotene content in various plant models by expressing the lycopene β-cyclase (LCYB) gene, which encodes the enzyme responsible for its production. In a study by Moreno et al. (2016), the expression of the LCYB gene from carrots (DCLCYB1) in tobacco led to an increase in pigment content, including β-carotene and chlorophylls, as well as enhanced levels of gibberellin (GA4) and plant biomass in T1 tobacco grown under controlled conditions [172]. This advancement highlights the potential utility of DCLCYB1 expression to promote growth and productivity in various crop species under field conditions. However, a comprehensive elucidation of the molecular mechanisms underpinning this phenotype is necessary for broader application [173]. Another study integrated multi-omics approaches, including transcriptomics, RNA-seq, proteomics, and metabolomics, to investigate the extensive impact of DCLCYB1 expression on the tobacco transcriptome and metabolic landscape [174]. The expression of DCLCYB1 led to substantial transcriptomic (~2000 genes), proteomic (~700 proteins), and metabolomic (26 metabolites) modifications, notably affecting pathways involved in cell wall biosynthesis, lipid metabolism, glycolysis, and secondary metabolism [175]. A gene and protein interaction network analysis revealed clusters primarily associated with ribosomal function, RNA processing, and translational activity. Additionally, genes and proteins related to abiotic stress responses were predominantly upregulated in the transgenic lines, aligning with observed enhancements in tolerance to high light intensity, saline conditions, and hydrogen peroxide-induced oxidative stress relative to wild-type plants [176]. These findings indicate a coordinated and systemic response extending beyond chloroplast-centric processes, encompassing nuclear and cytosolic components at the transcriptomic, proteomic, and metabolomic levels [177]. The results support the hypothesis that DCLCYB1 expression facilitates improved plant growth under optimal and stress-inducing conditions, underscoring its potential as a robust bioengineering target for developing high-performing crops [178].

Carotenoids are pigments with significant nutritional value in the human diet and are known for their antioxidant properties. These molecules act as free radical scavengers, boosting immunity and preventing cancer and cardiovascular diseases [179]. Alpha-carotene and beta-carotene, the predominant carotenoids in carrots (*Daucus carota*), also serve as precursors to vitamin A, whose deficiency can lead to night blindness and macular degeneration. To enhance the carotenoid content in fruit flesh, three pivotal genes from the carotenoid biosynthetic pathway were optimized for expression in apples [180]. These included phytoene synthases (DCPSY2) and lycopene cyclase (DCLCYB1) from carrots, as well as carotene desaturase (XDCRTI) from *Xanthophyllomyces dendrorhous* [178]. Each gene was cloned under the control of the *Solanum lycopersicum* (tomato) polygalacturonase (PG) fruit-specific promoter to ensure targeted expression in fruit tissues. The biotechnological approach was validated through subcellular localization studies and stable transformations in tomatoes (*Solanum lycopersicum* var. Micro-Tom) and transient transformations in Fuji apple fruit flesh (*Malus domestica*) [178]. The functional activity of the *S. lycopersicum* PG promoter was confirmed, successfully directing the expression of the transgene specifically towards the fruits. In transgenic tomato fruits, the expression of DCPSY2, DCLCYB1, and the combination DCPSY2+XDCRTI led to 1.34-, 2.0-, and 1.99-fold increases, respectively, in total carotenoid content compared with that in the wild-type fruits [178]. Additionally, the beta-carotene levels in tomatoes expressing DCLCYB1, DCPSY2+XDCRTI, and the triple combination DCPSY2+DCLCYB1 showed increases of 2.53-, 3.0-, and 2.57-fold, respectively, over those of the wild-types [173]. In the Fuji apple flesh infiltrated with DCPSY2+DCLCYB1 constructs, the total carotenoid levels increased by 2.75- and 3.11-fold, respectively, while beta-carotene content rose by 5.11- and 5.84-fold [180]. Although the expression of DCPSY2, XDCRTI, and the combination DCPSY2+XDCRTI+DCLCYB1 yielded more moderate yet significant changes in carotenoid profiles, these results confirm the efficacy of DCPSY2 and DCLCYB1 as promising biotechnological targets for enhancing carotenoid content in fruit tissues [173].

### 6.5. ZIPs and Its Homologs in Chloroplast Biogenesis

Chloroplast biogenesis is a fundamental process essential for photosynthesis, plant growth, and the production of key metabolites, among various regulatory factors [45,181]. Among these regulatory factors, ZIP transcription factors have been recognized for their role in chloroplast development, influencing gene expression linked to chloroplast formation. The EaZIP gene has gained attention for its potential involvement in chloroplast biogenesis within this family. Studies on a stable albino mutant of *Epipremnum aureum* “Golden Pothos” have provided insight into the genetic mechanism underlying chloroplast development, positioning the mutant as a promising model for functional analysis [181,182]. Albino mutant phenotypes were isolated from in vitro regenerated populations of variegated “Golden Pothos”, where the albino trait was previously attributed to the disrupted expression of EaZIP, a ZIP gene in *Epipremnum aureum*, encoding Mg protoporphyrin IX monomethyl ester cyclase, an enzyme for chlorophyll biosynthesis and chloroplast maintenance [183]. To further investigate the role of EaZIP in chloroplast biogenesis, researchers developed an efficient transformation protocol using petioles from mutant specimens as explants containing a traceable sGFP (super green fluorescent protein) marker. Notably, the expression of the *Arabidopsis thaliana* CHL27 (homolog of EaZIP), but not EaZIP itself, successfully restored green pigmentation and chloroplast formation in albino mutants, whereas EaZIP itself failed to achieve the same effect [184]. This suggests functional divergences between EaZIP and its homologs in chloroplast development. The regenerated populations exhibited phenotypic variations, including solid green, variegated, and pale-yellow foliage plants, demonstrating differences in chloroplast function and development. Thus, these studies underscore that these long-lived albino mutants, coupled with the optimized transformation system, offer a valuable platform for chloroplast biogenesis. Generating ornamental plants with diverse pigmentation facilitates research into physiological mechanisms linked to chlorophyll biosynthesis, chloroplast development, and other biological activities challenging to study in fully green plants.

### 6.6. Optimizing Photosynthesis with RuBisCO and CO₂ Concentrating Mechanisms

The heterologous synthesis of a biophysical CO_2_ concentrating mechanism (CCM) within plant chloroplasts holds considerable promise for enhancing the photosynthetic efficiency of C3 plants, potentially leading to significant improvements in crop yield [20]. In biophysical CCM systems, the mechanism effectively encapsulates RuBisCO, characterized by a high catalytic turnover rate, within an environment enriched in CO_2_ to optimize carboxylation efficiency [185]. A pivotal aspect of both naturally occurring biophysical CCMs and engineered analogs within C3 chloroplasts is the incorporation of functional bicarbonate (HCO_3_⁻) transporters and vectorial CO_2_-to-HCO_3_⁻ mass conversion enzymes [186]. Engineering efforts focus on the strategic localization of these transporters and conversion systems to the chloroplast of C3 plants, facilitating the elevation of HCO₃⁻ concentrations within the chloroplast stroma [187]. Despite identifying several CCM components from proteobacteria, cyanobacteria, and microalgae as promising candidates for this process, the successful functional expression of these components in C3 plant chloroplasts remains a challenge. This discussion outlines the complexities of expressing and regulating HCO_3_⁻ transporters and CO_2_-to-HCO_3_⁻ conversion enzymes within chloroplast membranes, which are critical for establishing a biophysical CCM in plant chloroplasts [188]. We address the extensive technical and physiological issues inherent in proposed engineering strategies and provide an overview of the current state of knowledge and existing knowledge gaps that must be addressed for successful implementation (Figure 4).

## 7. Technological Innovations in Chloroplast Research

Advancements in synthetic biology have emphasized the development of chloroplast genetic engineering as a promising alternative to nuclear genome engineering for crop improvement [189]. Chloroplast transformation, offering advantages such as biosafety through maternal inheritance and the ability to store specialized proteins, circumvents challenges such as transgene diffusion and protein toxicity. Following early breakthroughs in *Chlamydomonas reinhardtii* and *Nicotiana tabacum*, chloroplast transformation systems have been developed for over 20 flowering plant species, primarily dicots [190]. The efficiency of chloroplast genetic transformation depends on effective methods and well-designed transformation vectors, including selection markers and regulatory elements. Recent innovations have addressed the challenges, improved the transformation efficiency, and expanded the scope of this technology (Figure 5).

### 7.1. Genetic Transformations

Unlike Agrobacterium-mediated nuclear transformation, chloroplast genetic transformation predominantly relies on physical or chemical approaches. Among these approaches, biolistic bombardment remains the most widely employed technique for introducing foreign genes into recipient plants [191]. This method propels microprojectiles, typically tungsten or gold particles with diameters ranging from 0.6 to 1.6 µm, coated with the desired genetic material into chloroplasts under high-pressure helium within a vacuum-like environment [191]. While this approach is applicable across various plant species and cell types, the high-pressure penetration of microprojectiles can inevitably lead to cellular damage. Liu et al. (2024) found that the highest transformation efficiency (10.82%) and optimal result were achieved with four shots, 900 psi, and a bombardment distance of 6 cm. The microspore viability in the treatment group was slightly lower than that in the control group, while the embryo production and transformation outcomes were significantly enhanced [192]. This highlights the critical role of parameter optimization in maximizing genetic transformation success.

Polyethylene glycol (PEG)-mediated transformation has been utilized for chloroplast genetic modification. In this technique, plant protoplasts serve as the target cells, and exogenous DNA is translocated to the cell membrane, subsequently integrating into the chloroplast genome within vesicles under the influence of PEG [193]. However, this approach is significantly constrained by the low regeneration efficiency of the transformed protoplasts, which is highly species- and tissue-specific. Other methodologies such as glass bead-mediated transformation and microinjection have also been explored. However, their adoption has been limited due to challenges such as vector instability, narrow applicability across species, and suboptimal expression efficiency [194]. Consequently, the development of more efficient and versatile tools is imperative to enhance the scope and applicability of chloroplast genetic engineering.

### 7.2. Nanoparticle-Mediated Transfer

Nanomaterials have demonstrated revolutionary potential in advancing precise chloroplast transformation systems across various species due to their unique physiochemical properties [12]. As a cutting-edge tool in plant molecular research, nanotechnology offer numerous applications, including enhancing stress tolerance, improving material transport efficiency, increasing photosynthetic efficiency, and enabling the development of novel plant signaling molecules and pollutant detectors [195]. Ordinary single-walled carbon nanotubes (SWNTs) can penetrate the cell wall and membrane, but they lack precise localization in chloroplasts [196]. However, chitosan-wrapped SWNTs (CS-SWNTs) achieve targeted chloroplast localization at high surface charges. Their primary amines, at 6.5 pKa, allow for protonation under acidic conditions, enabling strong binding to plasmid DNA (pDNA) [197]. Chitosan, being biodegradable and non-toxic, ensures plant safety. Additionally, poly (acrylic acid) nanoceria (PNC) can target chloroplasts through a non-endocytic pathway, achieving a 46% co-localization rate [198]. A recent innovation in engineered nanotechnology employs a peptide recognition motif to deliver hydrophilic quantum dots (QDs) into chloroplasts. Encased in a β-cyclodextrin molecular basket, the QDs are small enough to pass through the cell wall pores of leaves [199]. By utilizing the highly conserved Rubisco small subunit 1A (RbcS) as a guide peptide in various species, this approach effectively overcomes biological barriers, thus facilitating targeted chemical delivery to chloroplasts.

### 7.3. Horizontal Chloroplast Genome Transfer

Genetic material is typically transferred between species through sexual reproduction, but recent studies have revealed that nuclear chloroplasts and mitochondrial genomes can also be horizontally transferred asexually between species. Chloroplast genome transfer can occur through grafting, where structural changes around the grafting site create larger stomata and transform chloroplasts into amoeba-like shapes to enable transfer between cells [200]. Researchers successfully transferred chloroplast genomes from transplastomic *Nicotiana tabacum* into *N. glauca* via grafting, yielding seven gene transfer lines. Six lines were homoplasmic, and one was heteroplasmic, demonstrating the feasibility of this approach for producing high-value compounds such as astaxanthin [201]. A novel technique called “cell grafting” has also been developed to horizontally transfer chloroplast or nuclear genomes using the callus cells of two parent plants.

Genome editing methods such as TALENs and CRISPR-Cas have significantly advanced genetic engineering. However, the double membranes of organelles in higher plants hinder the delivery of nucleic acids, limiting the application of CRISPR-Cas for editing plant organelle genomes [202]. Recent advancements, such as transcription activator-like (TAL) effector-based tools, have revolutionized base editing by enabling the precise targeting of organelle genomes through the modular assembly of TALE repeats. RNA editing in plant chloroplast genomes is crucial for their development and the accumulation of photosynthetic proteins [203]. Chloroplast-targeted base editors, such as the cytosine base editor (ptpTALECD) and DddA-derived cytosine base editor (DdCBE), allow for targeting C-to-T mutations while TAL effector-linked deaminases can induce heritable A-to-G mutations in chloroplast DNA, leading to phenotypic changes [204]. These tools are particularly effective in species where a chloroplast transformation system is not yet available, enabling modifications to chloroplast-encoded genes via nuclear transformation. Although CRISPR-Cas has shown potential in organelle genome editing, as demonstrated by its targeting of specific mitochondrial genome sites, challenges persist. A novel editing system designed for CRISPR editing of organelle genomes incorporates Cas9-type endonucleases, guide RNA, donor DNA, and a selection marker [205]. In the chloroplast genome of Chlamydomonas, this system facilitates precise donor DNA incorporation, relying on homology-directed DNA repair and replacement rather than generating indel mutations. Notably, Cas9 toxicity was absent in algal organelles, unlike earlier reports of nuclear Cas9 expression in Chlamydomonas. This advancement opens new possibilities for precise organelle DNA editing and the introduction of novel alleles without transgenic remnants [206]. However, challenges remain in enhancing editing efficiency and achieving homoplasmy in organelle genome editing (Figure 4).

### 7.4. Application of Genetic Transformation in Chloroplast

Chloroplast transformation has garnered significant interest from the pharmaceutical industry due to the high copy number of plastid DNA, which facilitate the large-scale production of vaccine proteins [207]. This approach involves integrating specific target genes into a plant’s chloroplast genome to produce bacterial or viral antigens responsible for particular diseases. These antigens are subsequently secreted as immunogenic proteins within the plant’s edible tissues [208]. For instance, LEKE, a promising target for blocking immune disease, has been successfully expressed in chloroplasts, yielding 6.3% of total soluble protein, or 2.3 milligrams of LEKE per gram of lean tissue. Moreover, recombinant protein accumulation was significantly enhanced by over 300-fold compared with nuclear transformation when the secretory human protein was expressed in chloroplasts [209]. Similarly, the HIV-1 p24 antigen was synthesized in a high-biomass tobacco cultivar, resulting in a significant increase in total soluble protein due to the addition of bacterial components [210]. This approach offers a promising platform for the production of vaccines and therapeutic proteins, leveraging the advantages of chloroplast transformation to achieve the high-level expression and accumulation of recombinant proteins.

Advancements in chloroplast transformation have been investigated to enhance nutritional and biochemical pathways, enabling the incorporation of essential nutrients into non-green plant parts. For example, a study on improving carotene biosynthesis in tomatoes demonstrated that the bacterial lycopene beta-cyclase facilitated the conversion of lycopene to beta-carotene, resulting in a four-fold increase in pro-vitamin A content within the fruit [211]. Additionally, lycopene beta-cyclase genes from Narcissus pseudonarcissus and Herbicola, when expressed in the tomato genome, enhanced the plant’s enzymatic activity to efficiently convert lycopene into pro-vitamin A beta-carotene [212]. Furthermore, the high-level expression of the GmTMT gene in transplastomic seeds significantly boosted the conversion of oil to tocopherols, achieving a ten-fold increase [213]. This improvement also enhanced the plant’s tolerance to salt and heavy metal stress by reducing reactive oxygen species, iron leakage, and lipid peroxidation [214]. These findings demonstrate the potential of chloroplast transformation to enhance the nutritional content and stress tolerance of crops, offering a promising approach for improving plant performance and human nutrition.

Over the years, scientists have extensively studied the roles of various genes involved in environmental stress responses. For instance, when the chloroplast genome of tobacco was engineered to express *Escherichia coli* L-aspartate alpha-decarboxylase, also known as aspA, which is responsible for catalyzing the decarboxylation of L-aspartate into beta-alanine and carbon dioxide, researchers found that the expression of *E. coli* aspA enhanced tolerance in photosynthesis and biomass production under high-temperature stress [13]. Oxidative stress, a detrimental factor for plants exposed to environmental challenges, has also been addressed through genetic modification. Researchers have improved the resistance of transgenic potatoes to oxidative and thermal stress by expressing superoxide dismutases and ascorbate peroxidases, which are key enzymes in mitigating oxidative damage [215]. These studies demonstrate the potential of genetic engineering to enhance plant tolerance to environmental stresses, such as high temperatures and oxidative stress, ultimately leading to improved crop resilience and productivity.

For example, sugars enhance the plant’s ability to eliminate reactive oxygen species (ROS) and stabilize macromolecules, thereby improving tolerance to salt, cold, and drought stress [216]. In transplastomic plants generated through somatic cell recombination and carrot cell transformation, the expression of certain genes significantly enhanced growth in high NaCl environments. These plants accumulated 50 to 54 times more betaine than unmodified cells cultured in liquid media with 100 millimolar NaCl [23]. This highlighted the role of betaine, which protects plants from oxidative stress, salt stress, and NaCl. Rice, with its ability to scavenge ROS, was also proposed to be able to enhance drought tolerance. Similarly, transgenic tobacco plants overexpressing the UV-CMO gene from Beta vulgaris, known for naturally synthesizing betaine, showed increased resistance to the accumulation of salt and drought stress [23]. These plants also demonstrated enhanced photosynthesis rates and yields under salt stress, specifically tested with 150 millimolar NaCl [13]. More recently, researchers reported that the expression of certain genes in transgenic plants enhanced their tolerance to various abiotic stresses, including salt, drought, and cold stress. These findings have significant implications for the development of crop plants that can thrive in challenging environmental conditions.

Transgenic chloroplast engineering has significantly enhanced plant growth and immunity under various stress conditions. *Nicotiana tabacum* (tobacco) expressing TPS1 (trehalose phosphate synthase) showed improved tolerance to drought and osmotic stress, enduring 24 days of drought and 6% PEG treatment [208]. Similarly, transgenic tobacco with the merAB operon facilitated phytoremediation under heavy metal stress, tolerating up to 400 μM phenyl-mercuric acetate and 300 μM HgCl_2_ [209]. Cold and chilling resistance was enhanced through the expression of Des (fatty acid desaturase), with leaf discs surviving 4 °C for 72 h and seedlings enduring 2 °C for 9 days [210]. Furthermore, CMO (choline monooxygenase) conferred resistance to toxic choline levels, salinity, and drought [211], while panD (aspartate decarboxylase) provided high-temperature stress tolerance up to 45 °C for 4 h [212]. Enhanced salt, chilling, and oxidative stress tolerance was achieved through the overexpression of DHAR, GST, and related enzymes, and oxidative resilience was further improved with Fld (flavodoxin) under 100 μM paraquat treatment [213,214]. Additionally, tocopherol biosynthesis pathway genes (HPT, TCY, and TMT) improved oxidative and cold stress tolerance at 4 °C for one month [215]. In Daucus carota (carrot), Badh (betaine-aldehyde dehydrogenase) enabled salt stress tolerance up to 400 mM NaCl [216], while Nicotiana benthamiana expressing sporamin, CeCPI, and chitinase demonstrated resilience to salt, osmotic, and oxidative stresses [217]. These examples underscore the potential of chloroplast transformation in addressing environmental challenges and boosting agricultural productivity (Table 2).

### 7.5. Challenges in Transplastomic Plant Development

While transplastomic plants have demonstrated outstanding potential in agricultural biotechnology, they have yet to fulfill initial expectations due to several technical, regulatory, and physiological challenges. One of the main limitations is the low transformation efficiency in many plant species, particularly monocots, as most successful chloroplast transition protocols have been developed for dicots such as *Nicotiana tabacum* and *Lycopersicon esculentum* [19]. Additionally, the stable expression of tRNA genes in chloroplasts is often hindered by unintended gene silencing, recombination events, and epigenetic modifications, which impact transgene expression over generations; furthermore, biocontainment issues related to horizontal gene transfer (HGT) remain a concern as chloroplast transformation is believed to be maternally inherited. Still, occasional paternal transmission has been reported in some species, raising concerns about gene escape into the environment [225,226]. Another major drawback is that transplastomic plants can still accumulate a high number of foreign proteins. The metabolic burden of overexpression may lead to reduced growth and fitness, particularly under natural environmental conditions [227]. Moreover, scalability and commercial adaptation are hindered by regulatory barriers and public perception, as genetically engineered crops continue to face strict approval processes in many countries. To overcome these challenges, future research should focus on enhancing transformation efficiency in monocots, optimizing transgene stability, and developing strategies to mitigate the potential ecological risk associated with chloroplast-engineered crops [228]. Addressing these key issues will help redirect the efforts of genetic engineers and biotechnologists toward resolving fundamental barriers and improving the feasibility of transplastomic plants for large-scale agricultural applications.

## 8. Future Research Directions

Future chloroplast research should focus on the critical integration of various internal signals, such as defense-related signaling molecules, and external signals, such as pathogen effectors and phytotoxins, to coordinate a synchronized whole-cell defense response during pathogen infection [3]. However, the precise mechanisms underlying chloroplast–pathogen interactions remain largely unexplored. Experimental validation should focus on localizing and characterizing chloroplast-targeted pathogen effectors, such as AVRPS4, HAPK1, HAPO111, HAPO12, and HAPM1, which are necessary to enhance our understanding of their roles in plant defense [229]. Additionally, identifying the direct chloroplast targets of these effectors can provide insights into how pathogens manipulate plastid functions during infection [229]. Another emerging area of research involves the effects of phytotoxins such as tabtoxin and phaseolotoxin on chloroplastic enzyme activities. These toxins are known to disrupt metabolic pathways within chloroplasts, yet their exact mechanisms of action and intracellular distribution remain poorly characterized [230]. Beyond plant defense, nanotechnology presents significant potential in advancing chloroplast research, including the targeted delivery of chemicals, the facilitation of precise gene editing, metabolic engineering, and real-time molecular monitoring [12]. Additionally, nanoparticle-mediated DNA delivery to chloroplasts is revolutionizing synthetic biology, enabling efficient transgene introduction across diverse photosynthetic species [12]. Developing nanosensors for detecting chloroplast biomolecules can enable the rapid detection of stress-induced metabolic shifts, while nanotherapeutics could be designed to enhance photosynthetic efficiency and stress resilience [12]. Merging synthetic biology and nanotechnology will further expand the potential of chloroplast transformation technologies [189]. The ability to engineer isolated chloroplasts with synthetic circuits could lead to self-repairing biomaterials that function using CO_2_, water, and sunlight. However, significant challenges remain, including understanding the precise mechanisms of nanoparticle entry into chloroplasts, improving chloroplast transformation efficiency, and translating lab-scale advancements into practical agricultural applications. Future research should develop in vivo and in vitro tools for the precise control of chloroplast function, transforming plants into bioengineered platforms for sustainable agriculture and environmental restoration.

## 9. Conclusions

Chloroplasts play a vital role in plant metabolism by regulating growth and defense trade-offs through photosynthesis, hormone biosynthesis, and stress signaling. This review highlights chloroplast-mediated reactive oxygen species (ROS) signaling, plastid-to-nucleus communication, and hormone synthesis (e.g., jasmonic acid, salicylic acid, and abscisic acid) as key regulators of plant adaptation under biotic and abiotic stresses. Recent advances in chloroplast genomics, transcriptomics, and engineering have provided deeper insights into the role of nuclear encoded transcription factors in chloroplast function, while synthetic biology and nanotechnology are emerging as promising tools for targeted metabolic engineering.

Recent technological breakthroughs, such as chloroplast-targeted gene editing (CRISPR-Cas9 and TALENs), horizontal genome transfer, and nanoparticle-mediated transformation, have significantly enhanced our ability to manipulate chloroplast functions for improved crop productivity. The expression of elongation factor EF-2 for heat tolerance and flavodiiron protein (FLV) for drought resistance and the metabolic engineering of carotenoid biosynthesis are concrete examples of home chloroplast modifications that can optimize stress tolerance while maintaining high yield. Additionally, multi-omics approaches (integrating transcriptomics, proteomics, and metabolomics) have enabled a system-level understanding of chloroplast metabolism, revealing novel pathways that can be harnessed for improved crop resilience.

Future studies should prioritize enhancing the efficiency of chloroplast transformation in non-model crops, as improving the disadvantages of current methodologies, such as their low transformation efficiency and homoplasmic conversion, will be crucial for broader application. Furthermore, integrating synthetic biology with AI-driven metabolic modeling could allow for the real-time monitoring of chloroplast gene expression and predictive analyses of plant response under stress conditions. A deeper investigation into chloroplast-to-nucleus signaling pathways may also provide novel strategies for fine-tuning growth–defense trade-offs, leading to enhanced stress resilience without compromising biomass production.

## Figures and Tables

**Figure 1 plants-14-00978-f001:**
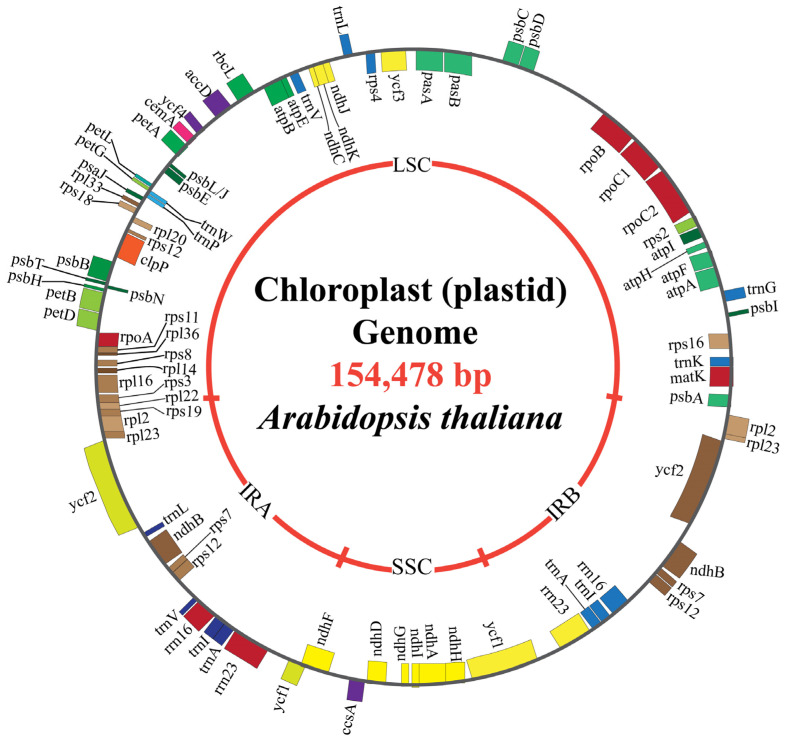
Circular representation of chloroplast (plastid) genome of *Arabidopsis thaliana*, spanning ~154 kbp, showing major elements including large-single-copy (LSC), small-single-copy (SSC), and inverted repeated (IRA and IRB) regions; various essential genes coding for tRNA and rRNA; and other important genes for plastid maintenance.

**Figure 2 plants-14-00978-f002:**
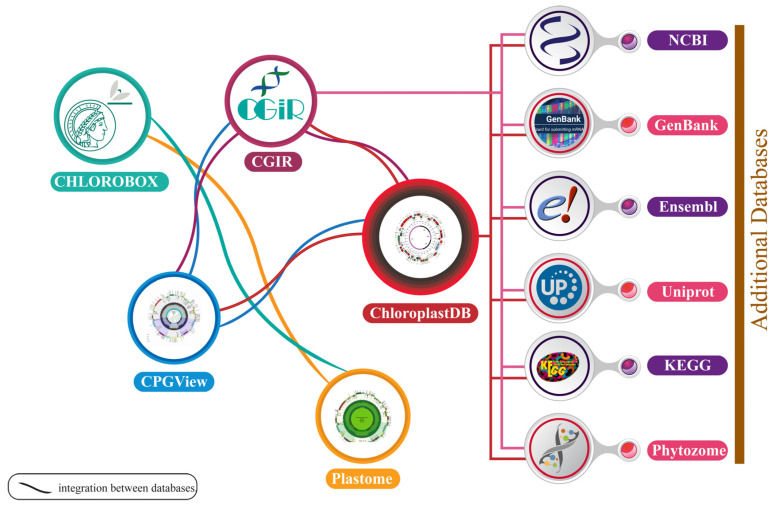
Illustration of the integration between various chloroplast genome databases and additional biological databases. The core database, ChloroplastDB, is interconnected with CHLOROBOX, CPGView, and Plastome, facilitated by CGIR. These databases integrate with primary external resources, including NCBI, GenBank, Ensembl, UniProt, KEGG, and Phytozome, enabling comprehensive genomics data retrieval, annotation, and functional analysis of chloroplast genomes.

**Figure 3 plants-14-00978-f003:**
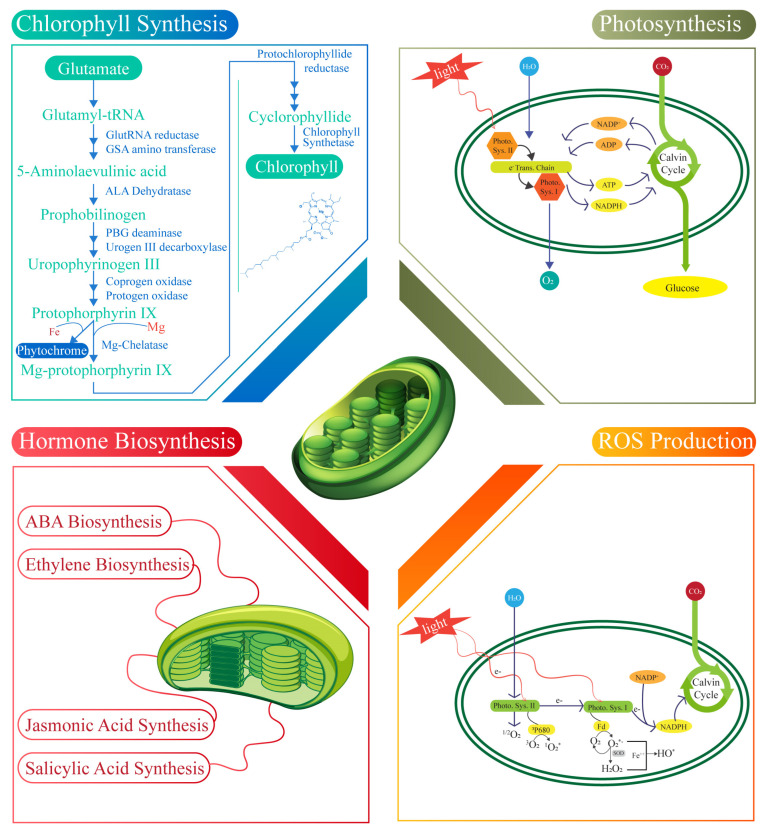
The multi-faceted role of chloroplasts in plants, including chlorophyll synthesis, through enzymatic pathways critical for photosynthesis; photosynthetic energy production, via light-dependent reactions; Calvin cycle hormone biosynthesis, essential for growth and stress response; and reactive oxygen species (ROS) production, which serves as double-edged sword in oxidative stress signaling during high light exposure. The interconnected process emphasizes chloroplast’s central role in balancing plant metabolism, growth, and defense mechanisms.

**Figure 4 plants-14-00978-f004:**
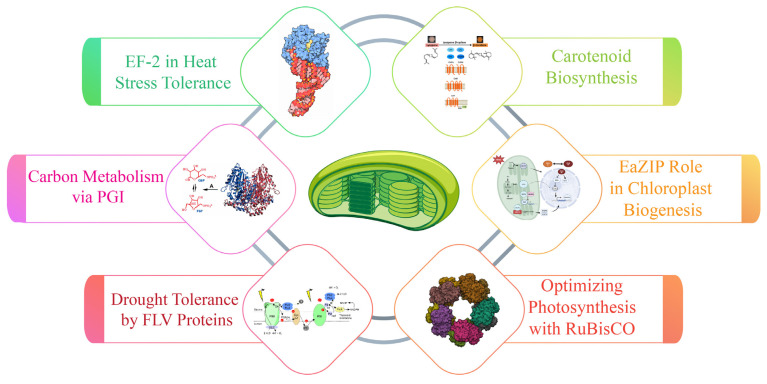
Outline of chloroplast’s role in plant growth and development, including photosynthesis optimization via RuBisCO, carotenoid biosynthesis for photoprotection, and chloroplast biogenesis regulated by proteins such as EaZIP. Additionally, chloroplasts play a critical role in carbon metabolism through PGI, while enhancing stress resilience via EF-2 for heat tolerance and FLV proteins for drought tolerance. Together, these roles highlight the importance of chloroplasts in regulating plant metabolism, growth, and adaptation to environmental challenges.

**Figure 5 plants-14-00978-f005:**
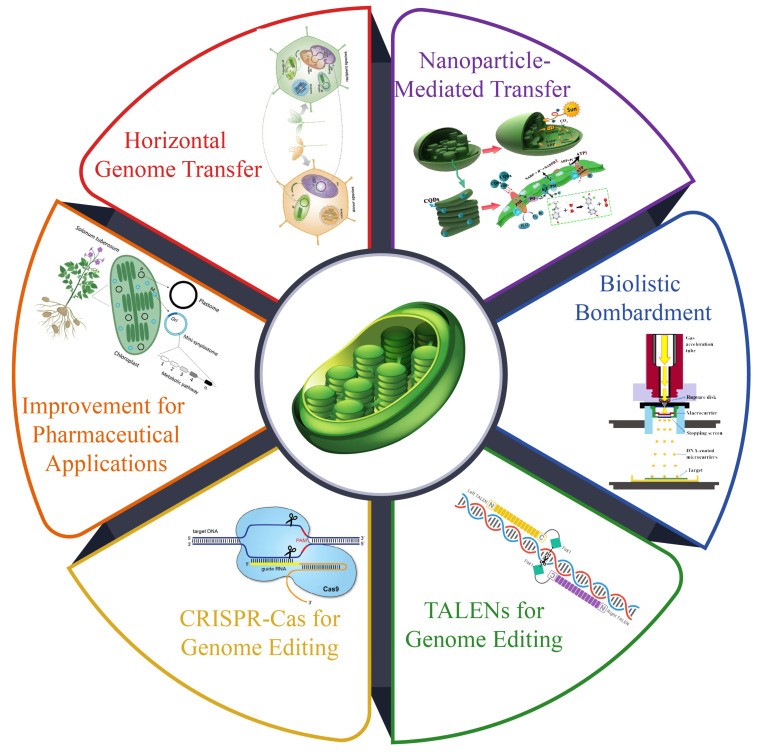
Innovative approaches in chloroplast genome engineering and applications, showcasing methods such as horizontal genome transfer, nanoparticle-mediated transfer, and biolistic bombardment for efficient gene delivery. Advanced genome-editing tools such as CRISPR-Cas and TALENs are highlighted for their precise genetic modifications, along with applications aimed at pharmaceutical advancements. Together, these strategies demonstrate the growing potential of chloroplast engineering in biotechnology and agriculture.

**Table 1 plants-14-00978-t001:** Chloroplast genome characteristics of various plant species, including genome size, presence and size of inverted repeats (IRs), single-copy regions (LSCs and SSCs), and number of genes.

Plant Species	Chloroplast Genome Size (bp)	Inverted Repeat (IR) Size (bp)	Single-Copy Regions (LSCs and SSCs)	Number of Genes	Reference
Tobacco (*Nicotiana tabacum*)	120,000 247,000	IR: 25,342	LSCs: 86,686; SSCs: 18,572	110–150 genes	[46]
*Asarum minus*	15,553	IR: 51,767	LSCs: 88,643; SSCs: 19,504	20–30 genes (reduced)	[47]
*Floydiella terrestris*	521,168	IR: 521,168	Not Annotated	~200 genes (largest cpDNA)	[48]
*Arabidopsis thaliana*	~154,000	IR: 26,263	LSCs: 84,197; SSCs: 17,780	131 genes (97 protein-coding)	[25]
*Lamiaceae species* (average)	149,081–152,312	2 IRs (with average 27%)	LSCs: ~82,000; SSCs: ~17,000	131–133 genes (86–88 protein-coding)	[32]
Maize *(Zea mays)*	140,387	IRa and IRb	LSCs: 82,355; SSCs: 12,536	120–130 genes (in some species)	[49]
Lima bean	150,902	26,543	LSCs: 80,218; SSCs: 17,598	130–140 genes	[34]
Cuscuta (parasitic plant)	96,292	None	None	<50 genes (photosynthetic loss)	[50]
*Zonal geranium*	217,942	26,000	LSCs: 130,000; SSCs: 25,000	200–210 genes	[51]
*Quercus ningangensis*	160,736	26,000	LSCs: 88,000; SSCs: 22,000	130–140 genes	[52]
*Sicyos angulatus*	154,986	IR: 26,276	LSCs: 84,355; SSCs: 18,079	130–140 genes	[38]
*Pistacia vera*	160,589	26,547	LSCs: 88,174; SSCs: 19,330	130–140 genes	[39]
*Sphaeropteris lepifera*	162,114	IRa, IRb: 24,028	LSCs: 86,327; SSCs: 27,731	140–150 genes	[40]
*Datura and Brugmansia*	154,686–155,979	26,500	LSCs: 82,000; SSCs: 17,000	130–150 genes	[41]
*Xanthium spinosum*	152,422	25,000	LSCs: 84,000; SSCs: 18,000	120–130 genes	[42]

**Table 2 plants-14-00978-t002:** Chloroplast transformation success stories. Stress tolerance in transgenic plant species: an overview of genetically modified plants, candidate transgenes, targeted stress, and stress conditions.

Plant Species	Transgene	Type of Stress	Tolerated Stress Treatment	References
*Nicotiana tabacum* (tobacco)	TPS1 (trehalose phosphate synthase)	Drought and osmotic stress	24 days of drought, 6% PEG	[217]
*Nicotiana tabacum* (tobacco)	merAB operon	Heavy metal stress (phytoremediation)	400 Î¼M phenyl-mercuric acetate, 300 Î¼M HgCl_2_	[22]
*Nicotiana tabacum* (tobacco)	Des (fatty acid desaturase)	Chilling and cold stress	Leaf discs at 4 Â°C for 72 h, seedlings at 2 Â°C for 9 days	[218]
*Nicotiana tabacum* (tobacco)	CMO (choline monooxygenase)	Toxic levels of choline, salt, and drought stress	30 mM choline, 150 mM NaCl, 300 mM mannitol	[16]
*Nicotiana tabacum* (tobacco)	panD (aspartate decarboxylase)	High-temperature stress	40 Â°C for 10 h/day for 1 week, or 45 Â°C for 4 h	[219]
*Nicotiana tabacum* (tobacco)	DHAR, GST, DHAR, gor, and GST gor (dehydroascorbate reductase)	Salt, chilling, and oxidative stress	200 mM NaCl, 12 days of germination at 15 Â°C, leaf discs at 8 Â°C for 8 h	[220]
*Nicotiana tabacum* (tobacco)	Fld (flavodoxin)	Oxidative stress	100 Î¼M paraquat up to 24 h	[221]
*Nicotiana tabacum* (tobacco)	HPT, TCY, TMT (tocopherol biosynthesis pathway)	Oxidative and cold stress	1 month at 4 Â°C	[222]
*Daucus carota* (carrot)	Badh (betaine-aldehyde dehydrogenase)	Salt stress	Maximum of 400 mM NaCl for 4 weeks	[223]
*Nicotiana benthamiana*	sporamin, CeCPI, chitinase	Salt, osmotic, and oxidative stress	Maximum of 400 mM NaCl, 3% PEG, 10 Î¼M paraquat	[224]

## Data Availability

All data utilized in this study are provided in the main manuscript, with relevant links also included.

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
