# Peer review of "Chloroplast Functionality at the Interface of Growth, Defense, and Genetic Innovation: A Multi-Omics and Technological Perspective"

_plants, 2025, doi:10.3390/plants14060978_

Round 1

Reviewer 1 Report

Comments and Suggestions for Authors

The manuscript will be of interest to a wide range of readers. It addresses a number of important issues. The manuscript is well written and illustrated. The list of sources is sufficient and includes many works from recent years. 

Recomendations

  1. The title of the manuscript should be clearer and more readable
  2. It is necessary to expand the list of keywords in accordance with the stated objectives
  3. Clearly articulate the novelty of this review study
  4. The Introduction contains too much general information that is not relevant. I would like the introduction to be logically structured and to highlight the further structure of the review, as well as to formulate the research question and its relevance.
  5. The Conclusion should be more specific information. So far, the Conclusion contains only general words. 

Reviewer 2 Report

Comments and Suggestions for Authors

The problem of obtaining commercially relevant transplastomic plants has been a long-standing one. However, the expectations of obtaining 40-60% of the target plant cell protein using this technology have not been met. Probably, serious changes in the technology are needed. At the same time, it should be noted that transplastomic plants were of great importance for theoretical work. In particular, they convincingly demonstrated the possibility of the existence of plastid RNA polymerase of nuclear coding.

A review will certainly be helpful. First, a very brief theoretical background is given, analysing the multifaceted metabolic role of chloroplasts, including even the involvement of chloroplasts in the synthesis of phytohormones and the involvement of phytohormones in the regulation of all plant life. Towards the end of the review, the authors bring the technology of transplastomic plant production more and more densely into focus, including methods of introducing exogenous DNA, new methods of gene modification are analysed, and so on. This review will not revolutionise the field, but it will be useful and that is what matters.

Since I am very familiar with the chloroplast genome, I hope that my advice will help the authors to make the review even better. The following are my recommendations. If there is a number in front it means the line to which my comment applies.

1) In part 2.1. it is highly desirable to provide a drawing of the plastid genome of, for example, tobacco or Arabidopsis or any other species that has a typical plastid genome. In this case, the data on genome size and described sequences will be much clearer for readers. There are a lot of such beautiful figures now in the literature.

2) In Table 1, the plastid DNA of the first 3 plant species does not show the size of IRs and LSC & SSC. These genomes are well known and it is better to cite all the data. The absence of these data is simply inexplicable.

3) The authors need to talk about the plastid genome where there is not a single gene I've been doing plastid genomes for many years, but I've never heard of such a thing. It seems to me that such plastid DNA does not make biological sense. The authors need to either clarify this point or remove it from the review. 

4) 163-164 and others. According to the authors, any positive or unusual property of chloroplasts or even genes leads to increased plant yield. Even the presence of multiple transcription initiation sites in the promoter of chloroplast genes is brought to yield by the authors. So far there is no reason to make such even a presumptive conclusion. Harvest is good, but it is meaningless to see harvest in every event.

5) 175 At the first stage of chloroplast development, NEP activity is necessary at least to establish the plastid translational system and to produce PEP that will go on to transcribe photosynthetic genes.

6) Since most chloroplast mRNAs are transcribed as large precursors, it would be useful to give at least some information on the processing of plastid transcripts. This is an extremely important step in plastid gene expression and absolutely must be considered when creating transplastomic plants.

7) As already mentioned, transplastome plants have so far failed to fulfil the hopes previously placed on them. It would therefore be extremely useful to analyse the reasons for this situation. For this review, this is absolutely necessary. Many genetic engineers and biotechnologists may then avoid going down the wrong path and focus on the most important problems in obtaining transplastomes.

The following are minor observations

123 between 2,500 and 2,500 proteins. Probably just a mistake

198 it would be useful to provide internet links to the databases discussed

342 “a plasma membrane-bound enzyme”.  Which enzyme are you talking about?

353 “to Novo”. Presumably the authors meant to write de novo?

159 Chloroplast genes are transcribed, not proteins.

434 PSI (photosystem one) and PSII (photosystem two). The transcription should be given at the beginning of the review, not at the end.

I wish you all the best               

Reviewer 3 Report

Comments and Suggestions for Authors

The manuscript covers an important and broad topic—chloroplast function in plant growth, defense, and genetic innovation. However, given the vast scope of the subject, the review lacks depth in key subtopics and appears somewhat scattered. Additionally, the accuracy of citations and the scientific discoveries summarized in the article need significant improvement. Below are specific comments for the authors to enhance the manuscript.

  1. The review lacks discussion on structural analyses, including the chloroplast protein translocon and its import motor, as well as recent advances in the structure of the chloroplast transcription machinery. Incorporating these aspects would strengthen the discussion.
  2. As a multi-omics summary, it would be beneficial to include advances in chloroplast envelope proteomics and recent findings on the chloroplast ubiquitinome.
  3. The potential of AI applications in chloroplast studies should be better elaborated.

Detailed points:

  1. Several claims throughout the manuscript require fact-checking, and some citations do not appropriately support the statements. For example:

Line 115: Verify the accuracy of the information.

Line 122: Citation (Ref. 31) seems inappropriate; please double-check both the citation and the accuracy of the statement.

Line 124: Ensure that Ref. 32 is complete and correctly formatted.

Lines 475-476: Ref. 127 does not mention the effector names listed; please verify and correct.

Line 538-540: The summary does not align with Ref. 145, which is not about chloroplast biogenesis or plastid-to-nucleus signalling.

Line 575 and Line 623: The authors use "we" when referring to studies they did not author (Refs. 153 and 163). Please revise for accuracy.

  1. Some sections are unclear, contain grammatical errors, or require rewording for precision:

Line 31: Should be "trade-off," not "trad-off."

Lines 38-39: "…supporting climate change…" is awkward; reword for clarity.

Lines 79-81: The sentence should be split for better readability.

Line 283: The phrase "Photosynthesis and chloroplast involve…" is unclear; please clarify.

Lines 546-548: Grammatically incorrect sentence—revise for clarity.

Line 606: "Engineering chloroplasts to incorporate FLV genes into the plant genome" is awkward, please revise.

Line 682: What is the "1" referring to? Please clarify.

  1. Figures and Tables

Figure 1: Correct "Unipot" to "Uniprot."

Figure 2: The stacking of PSII, ETR, and PSI in green shapes may misleadingly suggest a spatial organization. Consider revising the representation.

Figure 2 legend contains multiple typos: "multicfaced," "phtotsyncthesis," "dpeendedne," and "calvi." Please correct.

Table 1: The cpDNA copy numbers should be included, as they are referenced in the text.

  1. Additional Errors

Line 105: Missing word(s) in the sentence.

Line 123: "between 2.500 and 2.500" seems incorrect; please correct.

Line 253: Typo—should be "Chloroplast," not "Chorloroplat."

Line 263: "mon pyrrole"—please clarify.

Line 269: Should be "Mg²+."

Line 277: Should it be "chlorophyll q"? Please verify.

Lines 285-288:

    PSII should be "photosystem II," not "two."

    PSI should be "photosystem I," not "one."

    Cytochrome b6f should be "cytochrome b₆f," not "B6F."

Line 306: Likely meant "Shikanai," not "Chikinai."

Line 343: Clarify the meaning of "their elongation factors."

Line 412: "ICS2 double mutants…" Did the authors mean "ics1 and ics2"? Please check.

Line 426: Should be "5-10 mM," not "10-5 mM."

Line 502: What is CSK? It was not mentioned in Ref. 132.

Line 519: "Cyclotron" is unclear in this context—please clarify.

Line 922-924: The scientific basis for the proposed future direction (sustaining isolated chloroplasts) is unclear. Has this concept been tested? Please provide supporting evidence.

Lines 482, phaseolotoxins?

Please check the sentence in lines 352-354 to ensure it is correct.

In Section 6.1, Line 544. EF-2 operates in the cytoplasm, while chloroplasts use a different set of elongation factors. Please revise this section to focus more on chloroplast proteins (beyond the single sentence in Lines 552-553) or consider removing it.

Comments on the Quality of English Language

The English could be improved to more clearly express and articulate the points in the manuscript.

Round 2

Reviewer 2 Report

Comments and Suggestions for Authors

The authors have worked very carefully on the review, made a number of fundamental changes. The review has become much better. In this form it can be accepted for publication in the magazine.